# Assessing Institutional Stakeholders’ Perception and Limitations on Coping Strategies in Flooding Risk Management in West Africa

**DOI:** 10.3390/ijerph19116933

**Published:** 2022-06-06

**Authors:** Parfait K. Kouamé, Gilbert Fokou, Amoin Jeanne d’Arc Koffi, Amidou Sani, Bassirou Bonfoh, Kouassi Dongo

**Affiliations:** 1Centre Suisse de Recherches Scientifiques en Côte d’Ivoire (CSRS), Abidjan 01 BP 1303, Côte d’Ivoire; gilbert.fokou@csrs.ci (G.F.); jeannedarc.koffi@csrs.ci (A.J.d.K.); bassirou.bonfoh@csrs.ci (B.B.); kouassi.dongo@csrs.ci (K.D.); 2Centre de Formation en Santé Publique (CFSP), Lomé BP 917, Togo; saniamidou@yahoo.fr; 3UFR des Sciences de la Terre et des Ressources Minières, Université Félix Houphouët Boigny, Abidjan 01 BP V 34, Côte d’Ivoire

**Keywords:** coping strategies efficiency, floods, stakeholder analysis, transdisciplinary research, West Africa

## Abstract

Despite efforts at the national and international levels to mitigate adverse effects of climate change on the environment and human health in developing countries, there is still a paucity of data and information concerning stakeholder’s engagement and their level of collaboration, responses and assistance in West Africa. This study aimed at assessing the perception of institutional stakeholders and limitations on coping strategies in flooding risk management in Abidjan (Côte d’Ivoire) and Lomé (Togo). Using a transdisciplinary framework, the methodological approach basically relied on qualitative data collected through desk review and key informant interviews with various stakeholders, covering a range of topics related to flooding risk. Findings show that flooding experiences cause serious environmental and health problems to populations. Poor hygiene practices and contacts with contaminated water are the main causes of risks. Collaboration between stakeholders is limited, reducing the efficiency of planned interventions. Furthermore, health risk prevention strategies are still inadequately developed and implemented. Findings also show limited capacities of affected and displaced people to cope and plan for their activities. Engaging various stakeholders in the health risk prevention plans is likely to improve the efficiency of coping strategies in flooding risk management in West Africa.

## 1. Introduction

Flooding represents an important issue in Sub-Saharan Africa countries, especially in countries along the West African costal area. Floods associated with climate variabilities are experienced with the increase of temperatures, changes in the intensity of rainfall and extreme events [1]. Floods affect a large proportion of the people, disrupt livelihoods, destroy infrastructures, and reduce the food supply, as farmlands are often inundated [2]. In most West African countries, despite the development of management plans to deal with floods, the efficacy of interventions remains a big challenge [3]. In 2007, high rainfall caused severe floods, which affected more than 1.5 million inhabitants in West Africa. In 2012, flooding resulted in a death toll of 81 and 137 people in Niger and Nigeria, respectively, and the displacement of more than 600,000 people [4]. In Ghana, flooding experience has been chronic annually since 1990, with consequences on life, infrastructure and the environment and causing a death toll of over 150 persons, with severe damages of materials, such as houses in Accra [5]. Similarly, in Mali, especially at Yelimane in the Kayes Region, every year, large areas are flooded along riverbanks and lakes, destroying plant crops [6]. Despite the paucity of information on flooding in Togo, the few reports on flooding showed that in 2007, floods affected over 127,880 people, with 13,764 displaced individuals and 12 deaths [7]. In 2017, floods severely affected over 602 households from villages close to the Mono river in Togo [8]. In Côte d’Ivoire, projections confirmed the high increase of the intensity of rainfall combined with extreme precipitations [9]. Eight out of thirteen municipalities in Abidjan are at high risk of flooding [10]. Furthermore, during the raining season in June 2020, a deadly flooding occurred in several slums and residential areas of Abidjan. Flooding becomes a major threat to populations because of the poor and limited sanitation infrastructure, shortcomings in the urbanization, low capacity of local governments and limited coordination of relevant stakeholders [11]. Many factors are known as causing floods, such as climate variability and cumulative precipitations, anarchic urban sprawl and the poor urban planning of land use, with the increase of environmental vulnerability (e.g., water pollution, and impervious surfaces). Abundant inundation due to heavy precipitations and the overflow of water bodies often lead to critical socioeconomic, environmental as well as health impacts. Floods are accelerating the spread of water-borne diseases, causing environmental damages and degradation, and also well-being due to the poor intervention on sanitation infrastructures [12]. Apart from the environmental impacts, floods can lead to the collection of stagnant water, or solid waste, which provide breeding sites for mosquitoes. Further, rainwaters could reduce the risk of malaria by washing away breeding sites.

Due to the multifaceted impact of floods on environment and health, it becomes necessary to question the modalities to setup sustainable responses and resilient solutions. In such a context, the participation of various stakeholders in developing flooding management strategies is critically important [13]. The flood forecasting is an integral part of flood risk management through early warning systems. This is important to anticipate flood events [14]. In Nigeria, flood vulnerability maps were used for identifying at-risk populations, and setting up an early warning system and preventive measures [7]. In order to comprehensively analyse the sensitivity and adaptive capacity of some high value areas (e.g., world heritage sites) to climate change impacts, the bottom-up knowledge process is necessary [15]. To increase the societal resilience, it is therefore necessary to provide response to effects of climate change from below [15]. Floods management requires the identification and the involvement of various stakeholders, including communities and scientists with multilevel inputs and training of actors [16].

The complexity of the flooding pathways was highlighted in the context of Italy, where the planning of flooding interventions showed contradictions among actors, resulting in low adoption rates of interventions [17]. Some people hold the view that floods are not connected to climate change, even when there are recent flood victims [18]. Furthermore, local leaders of flood prone communities are not always included in flood risk management decision making processes [19]. Due to the failure of interventions, particularly in order to achieve the Sustainable Development Goals by 2030, an effective response is needed.

Climate change is known as a central theme of global sustainable development and specifically addressed in the Sustainable Development Goal (SDG) 13 [20]. Due to the complexity of climate change response and interventions, stakeholder’s engagement is necessary to reduce negative impacts by improving global, national and local interventions [17]. For SDG 13, multi-stakeholder contribution combined with infrastructural components should be integrated in programmes planning processes to increase their success. Preparedness of stakeholders and resilience of the health systems are necessary at community level in contributing to reduce exposure to risk and vulnerability [21]. To empower urban populations exposed to climate change, large-scale programmes were implemented for drainage infrastructures and national contingency plans. However, most of those initiatives used a top-down approach and were not adapted to local contexts, because of not involving the concerned stakeholders [22]. Consequently, these programmes failed because of the under-estimation of adaptation strategies of local stakeholders, and the low capacity to maintain large infrastructures in the long terms [23].

Serious gaps are limiting the achievement of global agendas for adaptation to climate change in West Africa. Efforts are noticed from governmental institutions and organizations working intensively to implement SDG13 interventions (e.g., infrastructure construction, policies, etc.), and to reduce the effects of climate change. Therefore, the under-estimation of adaptation strategies of local stakeholders remains, mainly coping strategies of populations, and local agencies. It is essential to question the relevance of existing responses, and weaknesses of the management strategies by featuring the knowledge of local stakeholders, in order to further inform the interventions, propose corrective measures, and to facilitate the uptake of research evidence in policy. With a transdisciplinary approach, the study aims to assess the perceptions of institutional stakeholders on coping strategies and limitations in flooding risk management in three municipalities in Abidjan (Côte d’Ivoire) and Lomé (Togo).

## 2. Materials and Methods

### 2.1. Study Area

As shown in Figure 1, this study was conducted in three municipalities in West Africa: Abobo and Cocody in Abidjan (Côte d’Ivoire) and Lomé (Togo). Abidjan is located in Southern Côte d’Ivoire between 5°10–5°30 North latitudes and 3°45–4°21 West longitudes. In Abidjan, annual precipitations range from 1500 to 2500 mm, with averages around 1784 mm, and the average temperature is 26.6 °C. Two municipalities in Abidjan that have been experiencing critical flooding risks for many years were selected, namely Abobo and Cocody. Abobo is the second most populated municipality in Abidjan hosting 1,030,658 inhabitants, while 447,055 people live in Cocody [24]. Cocody is a residential area in Abidjan while Abobo is more an unstructured and densely populated area. Even though the municipality of Cocody is well planned and structured, its high urbanization rate causes a deficit in sanitation infrastructures impairing evacuation of rainwater during flooding events. The main flood-risk factors in Cocody are the lack of adequate planning strategies for floods and effective coping mechanisms resulting in severe environmental damage. The municipality of Abobo is characterized by poor sanitation and economic conditions, a dense population and multiple shortcomings in solid waste and wastewater management (Figure 1). On the other side, the municipality of Lomé is the main local government of the capital city of Togo, with 837,437 inhabitants, according to the census held in 2010 [25]. The municipality is located in the extreme South-West of Togo, between 6°8′14″ North latitudes and 1°12′45″ East longitude. Since 2008, Lomé municipality experimented recurrent floods. The sanitation situation in Lomé is characterized by poor rainwater drainage management, with lack of drains, gutters without outlets, open gutters, and open pits used at 92%. That municipality has a raining water overflow drainage system comprising a retention basin network with 36 retention ponds. This area has a tropical climate, with two rainy seasons (March–April to July and September–November), and two dry seasons (August and November–February). The annual rainfall ranges from 800 to 900 mm. The average humidity exceeds 75%, where the monthly minimum values range from 57–58% during the dry season and 71–73% for the rainy season, while temperatures range from 23.1–30.7 °C [26].

### 2.2. Study Design

This study, which is a part of the Leading Integrated Research Programme for the Agenda 2030 in Africa (LIRA 2030, mainly uses a qualitative approach and relies on social science techniques for data collection and analysis. To better understand the stakeholder collaboration and current limitations on flooding risk response development in Côte d’Ivoire and Togo, the study employed a transdisciplinary approach (Figure 2). It required a stakeholder engagement process by conducting a workshop in Abidjan (Côte d’Ivoire) to explain the project vision and type of data to be collected in each involved country. Additionally, site visits were conducted in Abidjan to meet and to engage local stakeholders, prior to the beginning of data collection. One-on-one semi-structured interviews were conducted with key informants composed of representatives of agencies in charge of flood management. In Lomé, a one-day meeting was held with stakeholders to present the project and objectives, followed by secondary (desk review of reports and media documents) and primary data collection processes, especially for assessing flooding management and associated risks in Lomé.

### 2.3. Data Collection

Data were collected using several inquiry techniques: direct observation, semi-structured interviews, and participatory stakeholder workshops. Field observations were conducted in Abobo and Cocody in August 2017, and in Lomé, in April and May 2018, by trained research assistants under the close supervision of the research team. The objective was to gather information on wastewater drainage systems, response on flooding from municipal authorities, vulnerable areas and at-risk populations. To complement the site visits, unpublished reports, proceedings of workshops and meetings and media documents were reviewed to analyse the existing situation of flooding crisis in the study area.

A one-day participatory workshop was held on the 10 November 2016 in Abidjan with stakeholders from Côte d’Ivoire. Participants were selected based on their experience and involvement at various levels of the sanitation management system. The outputs from the workshop contributed to the design of the research proposal. This workshop contributed to assess achievements and failures of existing response strategies to flooding and disease control, and how the current research may help improving them. Overall, 35 stakeholders from waste management institutions, health and urban planning sectors, and municipalities were included. The workshop was based on a standard process and provided relevant information for data collection and to reinforce the engagement of stakeholders.

Based on a qualitative approach, the study used semi-structured interviews for data collection. One-on-one and face-to-face interviews were conducted with selected key-informants in Abidjan and Lomé, in July and August 2019, and from October to November 2019, respectively. The survey targeted several stakeholders involved in flood management, both in Côte d’Ivoire and Togo (Appendix A). An interview guide including open-ended questions was pre-tested in Abidjan and adjusted to the existing context prior to its implementation. In both Abidjan and Lomé, interviews were conducted in French by trained research assistants with experience in social sciences data collection. We conducted 20 semi-structured interviews with key stakeholders from governmental agencies on civil protection such as the National Agency for Civil Protection (ANPC), the National Solidarity Agency (ANC), the Center for Public Health Emergency Operations (COUSP); municipalities in Abobo, Cocody and Lomé; International organisations (UN agencies, such as the World Health Organization (WHO)), and civil society organisations, including mainly the Global Network for civil society organization for disaster reduction (GNDR). Practically, interviews lasted around 45 min and covered a variety of topics. These included urban flooding experiences, legal and institutional framework for the protection of the rights of vulnerable populations in case of flooding, and policy for the management of waterborne diseases during floods (Annex 1). Interviews were recorded using a digital voice-recorder and systematically transcribed in Microsoft Word. The transcripts were subsequently coded and analysed.

### 2.4. Data Management and Analysis

Transcripts of data from key-informant interviews were anonymized, and copies were stored on a hardware and the institutional server at CSRS. Data were then analysed using a qualitative data analysis approach. It consisted in the codification, categorisation, matching of data, presentation of results and checking of the data [27]. The approach used for textual analysis was Content Analysis (CA). CA can be viewed as an interpretation of the content of text data through the systematic classification process of coding and identifying themes or patterns. CA has the purpose to organize and elicit meaning from data collected and to draw realistic conclusions [28]. This method was particularly used for organizing ideas according to the convergence of meanings, establishing groupings between the respondents’ points of view on the theme of the study and drawing up a writing plan in order to interpret the results. The approach adopted in this study is the conventional content analysis, as compared to directed and summative approaches [29]. The particularity of the conventional approach to content analysis is the possibility to gain direct information from study participants without imposing preconceived categories or theoretical perspectives. The inductive approach of CA was applied, because it is more adequate for the situation with little literature supporting information, as observed in the case about flood management strategies in Abidjan and Togo. Data analysis of this study was conducted through the following steps: (i) reading of the verbatims of transcripts, (ii) identification of the key words and sections, and (iii) classification of highlighted sections according to the explanatory levels of the problem.

The transcripts from the interviews were manually coded based on a preliminary list of codes elaborated from the research objectives and completed during a careful reading of all the transcripts. Codes are textual labels in the form of keywords describing the content of an information, used to capture key thoughts or concepts. For this study, while a list of codes was validated by the research team before the beginning of the coding process, the coders were opened to new potential codes that could emerge from the transcripts. The coding was made by two different coders from the research team who are familiar with research questions and concepts. The benefit of using multiple coders is in the inclusion of multiple perspectives of researchers involved in the study and in the opportunities to discuss coding disagreements and refine the coding system [30]. Using multiple coders enhances the qualitative analysis and increases confidence in the results. Using the same codebook, all the transcripts were coded separately by each of the two coders based on their disciplinary backgrounds, skill levels and experience on the topic. Ref. [30] Coded sections from both coders were subsequently compared to assess similarities and differences in data. When the two coders disagreed on a specific section, they discussed until consensus and saturation of emerging issues was reached. Agreement through discussion at each step ensured that the results and their interpretation reflect a concordance of views between the two coders who, in this case, were also responsible for analysing and interpreting data.

Based on an inductive approach, labelled categories obtained from the initial scrutiny of data by the two coders were compared and contrasted to provide a framework for a thorough analysis of efficiency and limitations of institutional coping strategies in flooding risk management. The inductive approach is a qualitative method of CA used to develop theory and identify themes by studying documents and recordings. Data were classified, sorted, and arranged according to specific themes, mainly stakeholder types, flooding experiences, legal and institutional implications, assistance, and awareness messages, in order to develop meaningful conclusions. The aim was to define coherent meaning structures in the text material, taking into account the contextual (institutional and sociocultural) differences between Côte d’Ivoire and Togo. Seven main categories were defined as the main patterns of findings and presented in the results section. Those categories encompass: (1) characteristics and responsibilities of institutions involved in flooding risk management; (2) risks and disease transmission patterns; (3) collaboration between stakeholders in flooding risk management; (4) health aspect in flooding management policy; (5) institutional framework for protecting populations affected by flooding; (6) available capacities for displaced victims and assistance; and (7) implementation of awareness messages for preventing flooding risk.

### 2.5. Ethical Considerations

Before conducting the fieldwork in the Côte d’Ivoire, the project was approved by the national clearance committee in Côte d’Ivoire [112//MSHP/CNER/-km, IORG00075, 2017], because of the aspect involving mice capture and blood analyses in laboratory. For interviews in Côte d’Ivoire and Togo, prior to the study, informed consent was obtained from all individual participants, because of non-invasiveness and the low risk.

## 3. Results

### 3.1. Characteristics and Responsibilites of Institutions

Findings from discussions reveal that for the protection of populations against the recurrence of disasters due to human and natural factors, Côte d’Ivoire and Togo are equipped with an effective management system. As Table 1 and Table 2 show, there are several types of stakeholders involved in flooding risk management in Abidjan and Lomé with varying roles and responsibilities. In both cities, those actors range from governmental ministries and agencies to local governments (municipalities) international organisations (e.g., WHO), and civil society organisations (CSOs).

The review of legislative documents reveals that at the national level, the protection of the rights of vulnerable populations during disaster, including flooding, is a constitutional mandate of states. For example, in Togo, the constitution provides a legal and institutional framework for the prevention and management of disasters with a view to considerably reducing the loss of human life and material damage caused by the occurrence of hazards. In Côte d’Ivoire, disasters and emergencies situations are coordinated by the National Security Council (CNS), under the direct supervision of Head of State. This illustrates efforts from central government to coordinate activities related to disasters due to climate, pandemics, or socio-political unrest. This results in the creation of a National Platform for Risk and Disaster Reduction (PNRRC) in Togo, constituting a framework for synergy among actors and pooling of resources.

In Côte d’Ivoire and Togo, activities for flooding management are vested to several governmental departments and agencies with diverse and complementary responsibilities. In both countries, the main ministerial department involved in flooding management is the Ministry of Environment through the directorate of Health and Environment, and the Center for Public Health Emergency Operations (COUSP) in Côte d’Ivoire and Togo, respectively. Those entities provide health services and assistance to victims and homeless people during flooding events, and assess and monitor health risks. Other departments involved are the ministries of environment, infrastructure and construction, security and civil protection that are responsible, each in its domain, to contribute in the planning of waste management strategies and interventions, construction of infrastructure and monitoring of land use and planning, providing security and relief assistance during disasters. Ministerial departments are accompanied in the implementation of their tasks by several governmental agencies. The implementation of the disaster policy in relation to flooding consists in disaster risk assistance and evacuation of victims; collection and management of meteorological information; the building, control and surveillance of sanitation infrastructure; designing awareness messages for the media. In Côte d’Ivoire, those institutions include the National Office of Civil Protection (ONPC), the Meteorological operating company (SODEXAM), and the National Office of Sanitation and Drainage (ONAD). In Togo, in addition to the National Platform for Disaster Risk Reduction, the National Agency for Civil Protection (ANPC) and the National Solidarity Agency (ANC) play a critical role in that regard.

The second category of key stakeholders is the local governments composed of the municipalities of Cocody and Abobo (Côte d’Ivoire) and Lomé (Togo). Due to their proximity to communities, municipalities play a key role in the assistance to populations. They are vested to the design and implementation of a local multi-risk contingency plan for preparedness and responses to disasters. Their task consists in supervising urban land use, drainage of flooding areas, and the management retention basins for outflow of rainwater. They also assess needs and provide support in relocation and assistance to victims.

The last category of stakeholders is constituted of national and international Non-Governmental Organisations (NGOs) and other civil society organisations (CSOs) that provide support services to local institutions and communities in their efforts to manage the disaster. That support includes the provision of technical, financial and psychological assistance. They are, for example active in the implementation of health interventions (water, sanitation and hygiene programmes, provision of medication); provision of direct medical assistance, distribution of potable water and food packs to relieve flood victims. In the list of CSOs, international NGOs include mainly the International Committee of Red Cross (ICRC), Action Against Hunger, Save the Children, etc. During our interviews, participants also listed a range of local NGOs including Children of Africa, N’CLO BAKAN, NADO, Eau et Vie, Vie en Vert. Those organisations are working in collaboration with the National Office of Civil Protection (ONPC) in Côte d’Ivoire, and the National Solidarity Agency (ANC) in Togo. In this category, the role of traditional and religious authorities is also critical. Those NGOs collaborate with the authorities on an ad hoc basis.

Findings of this study on planning and organization aspects revealed limited interactions between stakeholders and integration of respective activities, absence of the planned budget for the flooding risk management system. Various entities work separately in their efforts to mobilise resources to contain the disaster when needed.

### 3.2. Understanding Collaboration between Stakeholders in Flooding Risk Management 

There are governmental and collaborative platforms to cope with flooding events and risk management in Côte d’Ivoire and Togo. For example, in Côte d’Ivoire, discussions during stakeholder workshops unveiled the existence of a formal collaboration between ministries, governmental agencies and municipalities to deal with flooding problems. The Ministry of Health together with the Ministry of Environment and Sustainable Development through the National Drainage Office (ONAD) work in collaboration with other governmental institutions, the ONPC and municipal authorities in Abidjan. A legal framework of collaboration exists between those institutions. This collaboration mainly concerns the provision of climate data and information on the variation of the weather by the governmental agency (SODEXAM). SODEXAM provides to scientists and decision makers with relevant information on climate and weather that is used to design awareness messages to prevent floods and consequences. Other collaboration types concern the dissemination of awareness messages: messages conceived by governmental agencies and technical services of local governments. These messages are disseminated through media, and especially used by religious and community leaders, neighbourhood representatives during awareness-raising activities in terms of flood prevention. Community and religious leaders, representatives of neighbourhoods are involved in various awareness-raising message dissemination for flood prevention. The study also showed the collaboration for the relief to population, and the post-disaster rebuilding.

In Togo, a similar framing of collaboration was observed with the aim to address the flooding risk between different institutions. As confirmed by an interviewee from a Togolese governmental agency, there is a formal platform for flooding and disaster risk management. It was noted in Togo that, “*according to the majority of organizations (ANPC, ANASAP, Platform of Disasters, COUSP and ANC), there is a coordinating body in the constitution of 14 October 1992, which constitutes a legal and institutional framework for the protection of the rights of vulnerable populations in the event of floods*” *(Interview with the responsible of the institution for civil protection, November 2019)*. We noticed a collaboration established between ministries, governmental agencies, technical services, local government, international and NGOs. This collaboration concerns the monitoring of stormwater drainage through COUSP, data collection, and the climate information sharing among the National Platform for Disaster Risk Reduction members. Discussions during interviews revealed that currently in Lomé, the flooding management system is being improved with the operationalisation of ANPC, with the support of the UN Office for the Coordination of Humanitarian Affairs. The collaboration also addressed the issue of supervision of land use, flooding risk areas through the local government, and the provision of assistance, post-disaster activities by several entities, Ministry of Health, and civil society organisations, such as relocation of displaced people and health assistance.

Based on the observed impacts of floods in the study area, the results revealed limitations in the efficacity of management platforms, and collaboration among actors, specifically on the level of this collaboration platform. Even though the expertise exists within governmental agencies, lack of financial resources is a major hindering factor for the implementation of various post-flooding interventions. Municipalities deplore the lack of resources (financial, capacity), reducing their prevention activities to awareness campaigns among populations living in at-risk areas. As indicated by some municipal authorities, in their interventions after a disaster, local governments basically rely on donations in kind (food and non-food items) received from the central government or other partners. Those partners are generally international institutions or NGOs with their own objectives, agendas, and resources, creating an asymmetry of powers between partners. As the coordinator of a local Ivorian NGO opined, “among partners, there are influential actors driving their own agenda and providing assistance based on their interests that could be political” (Interview with the coordinator of a local NGO in Abidjan, August 2019). This informant further indicated that an important local NGO in Côte d’Ivoire is coordinated by an influential political figure in the country, making disaster interventions less neutral. 

### 3.3. Flooding History, Risks and Disease Transmission Patterns

Results showed that both Abidjan and Lomé experienced flooding risks. In Côte d’Ivoire, flooding occurred since the last 30 years, as reported by respondents “*The ONPC has been working in the field of flooding for about 30 years*” *(Interview with ONPC representative, July 2019)*. Moreover, interviewees noticed an increase of flooding events over the past two decades in the selected municipalities in Abidjan. Participants explained that floods increased since 2009, with a high death toll, with an estimated average of 13 deaths each year, and affecting also many other cities, such as Gagnoa, Daloa, Bouaké, Agboville, San Pedro, and Korhogo. Semi-structured interviews conducted showed that Abidjan, as the biggest city in Côte d’Ivoire, is one of the main hotspots of flooding. Informal neighbourhoods were affected, because of the poor land use and limitations in the urbanization plans and sanitation infrastructures. Both formal and informal areas are concerned by floods posing the critical issue of landslide and death in Abidjan. The survey shows that all categories of population in Abidjan are exposed to floods, including poor people living in informal areas, such as Abobo. In the perception of the respondents, the most affected people are the most vulnerable segments of the population, including pregnant women, elders and children, homeless people, and street children. Consequently, people are exposed to waterborne diseases through contacts with contaminated flooding water, especially people with reduced mobility. The municipality of Cocody in Abidjan, composed of high standing residential areas, is surprisingly the most affected by floods. As the informant from ONPC analyses, “*this year [2019], it was noticed that the most affected municipality is Cocody, that is ironically a residential area populated by people with high living standings and benefiting from better road, drainage and sanitation infrastructures. In Cocody, 410 flooded houses, 26 partially destroyed and 4 totally destroyed houses were identified. The figures are completely different in the low-standing areas of Abobo where the floods did not reach that level of disaster*” *(Interview with ONPC representative, July 2019)*. Thus, contrary to Cocody, floods occur very often in Abobo, causing physical damages, but with mean incidence. Respondents have not mentioned any case of death due to flooding in that municipality.

Concerning flooding experiences in Lomé, “people are experiencing critical situations with flooding since 2008” *(Interview with informant from an institution for civil protection in Togo, October to November 2019)*. Furthermore, the survey in the setting highlighted that the main geographical areas affected by floods are the coastal zone and settings around the storm basins in Lomé municipality. The results in Lomé reveal that the increase of waterborne diseases transmission risk is related to many drivers, such as unsafe solid waste disposal, leakage from septic tanks or the poor management of wastewater from households. At-risk practices described by stakeholders included the non-compliance with the hygiene, the contact with contaminated water by the connection of the toilets to the water drainage system (e.g., the content of septic tanks dumped in the runoff), the open defecation, the open waste dumping, unremoved death animal corpses in the streets or the mismanagement of corpses of people who died during floods. Additional at-risk practices highlighted by informants are related to contaminations, such as the poorly constructed toilets close to wells, that could be a source of microbiological and chemical pollutant by exacerbating disease transmission during flooding events.

Interviewees revealed also, because of contamination of water supply system, the drinking water crisis during flooding is an important concern for the study area, when people do not have access to clean drinking water. For example, in Abidjan and Lomé, participants reported the risk of contamination of population by cholera or other intestinal diseases such as diarrhoea and dysentery, but also wound infections, skin diseases (e.g., scabies), and malaria caused by poor water quality and stagnant water providing breeding sites for mosquitoes. The survey, combined with grey literature revealed further practices (e.g., the release of wastewater from toilet in the environment or the open drainage and solid waste) that could contribute to the increase of the burden of water related diseases (e.g., skin diseases, diarrhoeas, cholera) during flooding events.

### 3.4. Analysis of Health Aspect in Flooding Management Policy and Institutional Framework for Protecting Populations

Discussions with representatives of involved institutions revealed the absence of a national strategical plan to deal with flooding related diseases prevention. The Ministry of Health and the Ministry of Environment and Sustainable Development are contributing to a global framework for disease management, that is not specific to flooding risk control. Nevertheless, WHO provided support to improving health condition to governmental structures. The District of Abidjan pointed out the necessity to integrate health risk aspect to flooding management strategies. Specifically, neither the municipalities of Abobo and Cocody, nor Lomé, do not have a proper policy for the management of waterborne diseases during floods. Although there is a service in the municipality to deal with health issues, for health equipment, infrastructure construction, donations of medicines, the strategic plan is only focused on prevention of landslide, and flood accidents due to people living in at-risk of flooding areas. As observed in Abidjan, the study showed that the municipality of Lomé has no measures for preventing health impacts of flooding events. Therefore, interviewees noticed that ANPC provides support for health, vector control and early detection of disease cases when needed, and trainings for the staff and the population. They contributed on hygiene aspects by providing chlorine tablets for the treatment of drinking water, in collaboration with COUSP that performed the treatment of water points, provided chlorine for household water treatment and raised awareness about diarrheal cases.

In the study area, participants revealed the existence of documents on legal framework for flood management revealed that the National Plan for disaster Management (ORSEC Plan) in each country is implemented under the governmental body, through the Prime Ministry, and other contributions, such as the Ministry of Health, the Ministry of Environment and the Sustainable Development, the Ministry of Defence, and ONPC. It appears from the discussions this legal and institutional framework for protecting the rights of populations in case of natural disasters is poorly understood by practitioners. In the municipalities of Abidjan, the ORSEC Plan is managed by the mayors, as they are the main actors and responsible for disaster risk management. Similarly, in Lomé, discussions revealed that the legal status of flooding risk management about the ORSEC Plan, that is under the coordinating body in the Constitution of 14 October 1992, the legal and institutional framework for protecting the rights of vulnerable populations. Additional laws and decrees contributed to disaster risk management, and for protecting people. Further framework law and decrees were used for the environment risk monitoring, the disaster relief organization plan (ORSEC-Togo), the law on the organization of public services of the drinking water and the collective sanitation of the domestic sewage, and the Disaster Platform.

Despite the existing institutional framework for the protection of populations against floods, some limitations persist. As indicated by the Ivorian Ministry of Health some major actors lack the mandate in some contexts to fully implement planned interventions. As discussions revealed, “*the Health and Environment Directorate* [of the Ministry of Health] *has the legitimate task to implement floods preventive measures. This consists of designing communication plans, awareness-raising and educating populations by showing them the risk of staying in unsafe places in the event of flooding. However, we do not have the capacity and the right to expel people from those areas*” (Interview with informant from Ministry of Health, Abidjan, July 2019). Those actors plead for more collaboration between health, security, environment and sanitation, and humanitarian sectors. 

### 3.5. Available Capacities for Displaced Victims and Assistance

Difficulties in programming in advance capacities for displaced victims in the study area were raised in Abidjan. This is due to some constraints associated with land use, availability and the random occurrence of floods. Furthermore, other explanations were raised concerning the sites location for flooding related-displaced victims that should followed some criteria: the relocation site not far from the home of victims; the site should not be flooded in case of rain; and the space must be large enough, because of the children, under the supervision of police forces and the ONPC. Discussions with ONPC in Abidjan indicated that despite the absence of a specific sites for victims, public schools, stadiums, and churches could be used for displaced persons, when needed during disasters events, including flooding victim’s relocation. Additional aid could be provided by NGOs for assistance to affected people when they are already settled in a specific site. In the case of Abidjan, interviews showed that ONGs built the first aid alert stations to take care of small disease cases, with the contribution of the General Direction of the Health in Côte d’Ivoire. Victims received psychological assistance under the supervision of medical healthcare centres.

As for Lomé, the study highlighted that due to critical status of occurrences of floods, sites designed for receiving displaced victims are identified in advance for different capacities, with the support the army forces. The disaster management platform checked sites availability, the closest to the disaster area, not flooded, and the security conditions. For the concerns of assistance medical care were addressed to support populations. Other assistance comprised drinking water availability and quality for victims. Interviews revealed also that ANC contributed to the identification of the victims, the distribution of foods, and equipment to victims.

In both countries, assistance to displaced victims during flooding is often uncoordinated, and expose populations to more risks. As an informant describes, “after flooding events in Abidjan, a company was contracted to build improved shelters for victims. However, there was a problem with those shelters because they were not compartmentalized. It was a large hall where the comfort is not guaranteed. Worse, there were risks of epidemics because if someone has tuberculosis, it could be complicated” (Interview with the coordinator of a local NGO in Abidjan, August 2019). Adding to that, relocation sites for victims are selected without consulting neighbouring populations who reluctantly accept the presence of victims in their close vicinity for security reasons.

### 3.6. Implementation of Awareness Messages for Preventing Flooding Risk in the Study Area

Stakeholders in charge of flooding risk prevention delivered awareness information for raining seasons the population through different channels (e.g., SMS via telephones, TV and radio messages, and Internet). In the study area, governmental departments provided flooding information through meteorological agencies. In Côte d’Ivoire SODEXAM provided regularly these messages, when Ministry of Health and ONPC advised and raised awareness through messages on risk faced by people living in flood-prone areas. Additional awareness messages were addressed to illiterate people in several languages with the support of local radio and community leaders to reach a large number of people. The same framework for community outreach was observed in Lomé. It contributes to the sensitisation of city dwellers through the Risk and Disaster Reduction platform in charge of early warning system. For example, participants handed over messages on taking refuge, good practices on sanitation and drink water quality, hand washing before eating, cooking, after defecation, before breastfeeding, when needed immediately take the patient to healthcare centre. Furthermore, to address needs and expectations of populations, messages are elaborated before translating into local languages and transmitting by endogenous animators.

## 4. Discussion

This investigation assessed the institutional stakeholders’ perception on coping strategies and limitations in flooding risk management in two cities in West African Countries: Abidjan in Côte d’Ivoire and Lomé in Togo. The comparative assessment showed that the flooding risk management integrated a various type of stakeholders, with mostly some similarities on flooding experiences, the existence of legal and institutional framework, the management of waterborne diseases, and collaboration, capacities available or not for victims and assistance. This investigation contributed to understand the efficiency, limitations, knowledge gap for developing global policies to cope with the issue of resilience to climate changes impacts in West Africa.

### 4.1. Experiences of Flooding and Waterborne Diseases Transmission Patterns

This investigation conducted in Côte d’Ivoire and Togo showed similar flooding management structures comprising the Disaster Management Platform, the National Protection Agency, the Ministry of Health, and National agencies, such as the National Office of sanitation, the National Civil Protection, with the support of national and international NGOs and multilateral agencies. As presented, this architecture of flooding risk management including multi-level implication for a sustainable decision making seems to be well-established to handle the exposure to flooding events in the study area.

Therefore, since some decades, the selected municipalities, including Abobo and Cocody (Abidjan) and Lomé (Togo) are still exposed to severe flooding risks. Indeed, due to climate changes, flooding experiences represent a critical issue worldwide, and specifically for Sub-Saharan African countries. A recent systematic review unveiled an increase of climate risk in Africa and Asia showing the difficulty for the adoption of climate adaptation measures [31]. Exposed areas to flooding risk were shown near the Ouladine lagoon and the Ebrié lagoon close to Comoé river in Côte d’Ivoire [32]. This finding confirmed the similarities of flooding impacts in West Africa, posing the critical challenge of developing sustainable management strategies by governments in these countries. A study conducted in Accra (Ghana) confirmed this observation by presenting rapid urbanisation and changing climate as drivers that complicated the frequencies, intensities, and associated impacts of disasters [33]. Authors argued that the conventional approaches overly dependent on strengthening technical and physical infrastructure are inadequate [33]. In climate change context, the settlement expansion into food-prone areas is expected to increase food risk in the future in West Africa. Infrastructural measures were a dominant category of measures before and after food events [34].

Floods posed serious impacts affecting houses in terms of destruction, and physical damages in the study area. A report from ONPC indicated that in 2018, floods caused 410 flooded houses, 26 houses partially destroyed, and four houses totally destroyed in Abidjan. Climate change is considered as a primary global driver of migration, due to extreme weather events such as storms, and floods [35]. Though the observed situation of flooding in our study was not related to the outflow of the sea level, it is necessary to develop mitigation measures. A study argued that Côte d’Ivoire, Ghana, and Togo the major drivers of environmental degradation are originating from rapid, inadequately planned and managed urban development [36]. Furthermore, these authors valued the impacts of degradation resulting from flooding and erosion, at over US $3 billion by 2100, based on the worst-case scenario of regional relative sea-level rise. Our study showed the need of research to contribute to in-depth assessment for improving official climate data, impacts, and their availability to plan strategies and sustainable decision making. A study conducted in India addressed the necessity of data ability at the watershed scale to simulate flood events, and to predict flood-prone areas, considering multiple rain gauge data, that could facilitate more accurate flood inundation [37].

The non-compliance with hygiene practices, the contact with the contaminated water by the connection of the toilets on the pipes and the content of septic tanks dumped in the runoff, open defecation, and open waste dumps are key issues exacerbating the risk to diseases. Flooding events are more often associated with diseases, where sanitation and waste management system are limited. For example, a study conducted in Uganda showed the presence of rotavirus concentrations in rivers with key sources from open defecation, population growth, urbanization, limited treatment of wastewater and poor faecal sludge disposal and poor sanitation coverage [38]. In Accra (Ghana), for instance, due to the lack of proper drainage, sewage and other refuse, effluents were dumped into the sea and gutters. There are few reports on waterborne disease contamination related to flooding events in Togo. Therefore, a study conducted for examining water security in the challenging environment in Togo revealed inadequacies in water quality, and the absence of de-centralized water management structures [39]. As presented by the authors, this situation of water could render difficult interventions during flooding periods in this area.

### 4.2. Collaboration between Stakeholders in Flooding Risk Management

To deal with environmental and health impacts of flooding in the study area, integrated efforts are required for management platforms, and among actors. Therefore, the results showed the collaboration among stakeholders is not well established to increase the efficiency of planned interventions, both in Abidjan and Lomé. Flood risk management requires participatory governance, to the analysis of potential solutions, the management of consequences, consideration of groups’ interests and values, and better strategies for communication to victims [40]. These observed difficulties in collaboration for flooding risk management is not specific to the study area. Research conducted in the UK showed that stakeholders’ participation to the integrated team for flooding risk management come with challenges, such as limitation of financial resources, stakeholder’s spatial distribution, and their interest to participate [41]. Another research investigated how governments, non-governmental organizations and at-risk communities perceive flood risk and collaborate in flood risk management decision-making processes. They revealed that divergent perceptions of flood risk between at-risk communities and the governments impede realization of flood risk reduction goals [42].

### 4.3. Understanding the Integration of Health Risk Aspect in Flooding Management Policy in the Study Area

Our assessment showed that the integration of the policy of health risk prevention and management when planning flooding mitigation strategies is limited in Côte d’Ivoire and Togo. Balancing ecosystem and health aspects is relevant for achieving the goal of sustainable urban planning. For example, a higher temperature in urban areas could exacerbate the health risk [43]. Ecological systems are complex with several issues that are not integrated in the social governance system. Notable advances have been observed in some countries such as Switzerland. A recent study conducted in this country showed that for flood risk management, actor- and law-based issue integration co-vary and might be self-reinforcing [44]. The issue of integration rested on laws, although cases exist where actors are the main basis of integration [44]. Furthermore, as observed in the case of our study, sometimes, though flooding risk framework was integrated, the social dimension was not addressed, as result the inefficiency of the mitigating strategy. For example, a multi-layer safety approach was introduced in Netherlands from flood prevention to flood risk management, by integrating three components, such as defensive measures against floods, resilient spatial planning measures, and effective disaster management measures [45]. Another investigation conducted in Sweden argued that for assessing flood risks, it is necessary to evaluate measures, and to reinforce engagement of stakeholders [46].

### 4.4. Analysis of Legal and Institutional Framework for Protecting Vulnerable Populations during Flooding Events

For the concerns of legal and institutional framework for flooding risk mitigation, our study revealed the availability of ORSEC Plan for disaster Management in each country supervised by the governmental body. However, this strategical plan is not well promoted and know by practitioners. Some reasons could explain the non-application of ORSEC Plan for the flooding risk management, such it includes high level authorities, implemented for top disaster occurrences. Therefore, as observed in our study, and due to occurrences of floods, it is necessary to provide specific law and stakeholders to address this issue. For the study area, the legal and institutional framework could be more realistic and practical addressing population protection, displacements, prepare to changes, and assistances (foods, medicals, etc.). For example, a study realized in EU countries in response to flooding risk provided recommendations, such as prepare for changes by developing multi-functional and flexible plans, and make space for innovation by seeking to manage risk rather than avoiding it [47]. Yet, there are additional laws to the ORSEC Plan in each country. As diseases occurred after flooding events, these regulations should be reinforced to improve flooding risk, for changing existing practices of flood risk management at a local level, as recommended some authors [48]. A study conducted in China highlighted that the institutional framework could address the gaps about conflicts through massive relocation, the financial limitations that could affect the policy feasibility [49]. Furthermore, other implications could be relevant, such as the private sector, and community organizations, such as the nonstate actors that played an important role in post-disaster aid [50].

### 4.5. Status of Available Capacities for Displaced Victims, Assistance, and Awareness Messages

Difficulties are noticed for the planning in advance capacities or sites for displaced victims are observed in the study area. These could be associated with some constraints, such as the shelter should not be far from the home of victims, the security of victims, and the need for large space for children. Though the management of flooding risk requires the planning in advance of shelter is difficult for most of countries, because of the poor urbanization, land use, and the population growth. The availability of capacities is a challenge worldwide, in the critical context of climate changes associated with floods. A recent investigation in Bangladesh, that experienced yearly flooding through cyclones, floods, and riverbank erosion provided a strategical 5-year plan to understand the governance of displaced population [51]. This planning of governance of displacements of victims is an innovation and could be relevant for our study area in West Africa, for the better follow-up of social and health impacts of flooding event on victims. Furthermore, a study conducted in Zimbabwe analysed the gap in political attention as a root to address internal displacement, by showing the lack of social and economic as contributing to exacerbate the plight of displaced persons [52]. Furthermore, in the study area in Côte d’Ivoire and Togo, different forms of assistance were provided to victims in terms medical care, clean drinking water, and foods.

This situation is similar as observed in cases as post-flooding response worldwide, and specifically in Sub-Saharan Africa. For instance, in Indonesia, to deal with floods and impacts, a collaborative endeavour of multiple stakeholders was implemented by strengthening social capital [52]. To cope with flooding related risk, in the study area, awareness information was provided to prevent populations for raining seasons through different channels, where some are translated into local languages. This finding is relevant to save life in the context of the poor urbanization and low access to sanitation infrastructure for the sustainable management of flooding risk. Therefore, awareness raising should be integrated into an holistic framework or a general agenda, for reducing risk to ensure sustainable and productive decision making. Awareness raising could vary from a group of residents characterized by low-risk awareness and high trust in structural flood protection. It depends on living in urban or rural settings, communities characterized by a lack of or limited experience with floods, and required strong implications for local risk managers [53]. Authors suggested due to the challenge of the volume, variety, and the veracity of information, it is realistic to use social media messages, and render them in an easily comprehensible format for various stakeholders to gain deeper insights [54].

### 4.6. Strengths and Limitations of the Study

This study has several limitations. Firstly, it was conducted within a collaborative project, named “Leading integrated research for Agenda 2030, LIRA2030”, and we had some delays on data collection due to some rearrangements. Fieldwork in Togo was conducted one year after collecting data in Côte d’Ivoire. Though the research was conducted during raining seasons in each country, this limitation could introduce some biases in data analyses in the study area, because of stakeholder availability. Secondly, we were unable to assess the level of collaboration through the interview survey with participants in the study. To improve the quality of results about perceptions, collaborations, and for a comprehensive legal and institutional framework by practitioners, focus group discussions with communities might be helpful to determine the most insights, limitations and how to propose further response to the situation. It would have been interesting to also integrate other stakeholders, such as populations most at risk in this study, because the results of this qualitative research are only based on the knowledge of institutional stakeholders. Finally, our results from qualitative data did not make it possible to deeply assess the link between SDG 13 and SDG 6, that could be an added value to this investigation. Further research could address this critical issue in urban settings in West Africa, in order to provide new evidence for decision-making.

## 5. Conclusions

This assessment of the perception of institutional stakeholders on coping strategies in flooding risk management showed that risk of floods is critical in the municipalities of Abobo and Cocody in Côte d’Ivoire and Lomé in Togo, as a consequence of climate change and the few collaboration between stakeholders. The improvement of institutional collaborations could increase the efficiency of implementation of interventions for displaced victims, and then reduce the risk of diseases transmission. Furthermore, there is a lack of integration of health risk aspect for the planning mitigation strategies. There is a legal and institutional framework for disaster management, specifically for flooding risk mitigation. Therefore, its implementation encounters difficulties due high-level authorities’ implication. Flood risk management requires participatory governance, that is critical in defining potential solutions, the management of consequences, consideration of interests of various groups and their values, and better strategies for engaging and communicating with stakeholders. The study recommends specific regulations for addressing the issue of flooding occurrences observed each year in the study area. Difficulties are noticed for the planning in advance capacities or sites for displaced victims are observed in the study. Consequently, limitations are noticed for the availability of capacities for displaced victims. It is necessary to early planning the governance of displacements, that is essential for a better follow-up of social, environmental and health impacts of flooding events in West Africa.

## Figures and Tables

**Figure 1 ijerph-19-06933-f001:**
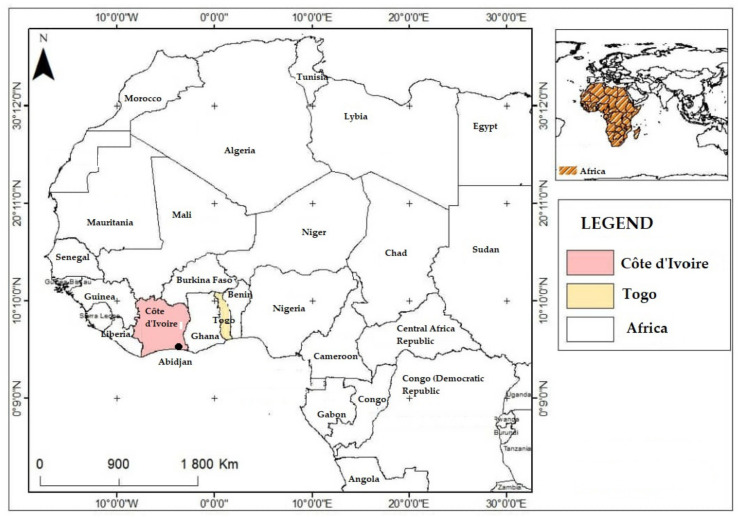
Study area location: Abidjan (Côte d’Ivoire); and Lomé municipality (Togo). Reference: www.divas-gis.org, accessed on 5 March 2022.

**Figure 2 ijerph-19-06933-f002:**
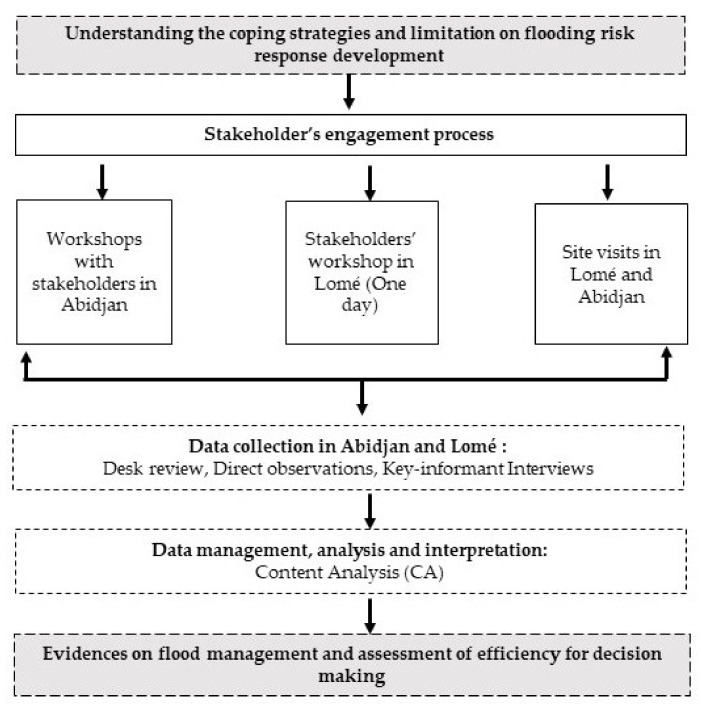
Research framework for understanding data collection process in the study.

**Table 1 ijerph-19-06933-t001:** Stakeholders and implication in flooding risk management in Abidjan.

Type of Stakeholders	Stakeholders	Role in Flooding Risk Management
Abidjan, Côte d’Ivoire		
Ministry departments	Ministry of Health	• Disease control and prevention, health risk surveillance and monitoring
	Ministry of Environment and Sustainable Development	• Planning of waste management strategies and interventions
	Ministry of Infrastructure and Construction	Construction of infrastructures Monitoring of land use and planning
		• Providing assistance during disasters (e.g., floods, fire, etc.)
	Ministry of Home Affairs/Security and civil protection *	• Providing security and relief assistance
	National Office of Civil Protection (ONPC)	• Disaster risk assistance and evacuation of victims
Governmental agencies		Collection of meteorological information Provide alerts for flooding risk prevention Sharing data with governmental agencies and scientists
	Meteorological operating company (SODEXAM)	

	National Office of Sanitation and Drainage	Building of sanitation infrastructures under the supervision on the Ministry of Infrastructure and Construction, and the Ministry of Sustainable Development and Environment Control and surveillance of sanitation infrastructure
Local Government	Municipalities/Abidjan District	Supervision of land use, drainage, flooding areas, and the management retention basins for outflow of rain reduction. Providing support in relocation and assistance to victims.
Civil society organisations	National NGOs: N’CLO BAKAN, NADO, Eau et Vie, Vie en Vert International NGOs: Red-cross, Children of Africa	Support to the National Protection agency (ONPC) in the implementation of intervention (WASH, health assistance to victims) Providing financial assistance to victims under the supervision of the municipality. Providing food packs, medication, psychological assistance, with support of ONPC for disaster intervention (Medical assistance and WASH). Donations to relieve flood victims

* These are currently two separate ministries, but they formed a single department at the time of the field study.

**Table 2 ijerph-19-06933-t002:** Stakeholders and implication in flooding risk management in Lomé.

Type of Stakeholders	Stakeholders	Role in Flooding Risk Management
Lomé Municipality, Togo		
Ministry Departments	Ministry of Health	• Providing health services and assistance to victims and homelessness people during flooding events
	Center for Public Health Emergency Operations (COUSP)	• Providing health assistance under the supervision of the Ministry of Health
	Ministry of urbanism and Habitat	Monitoring stormwater drainage system, protecting flooded areas, implementing laws and regulation on urban land management Ensuring the cleaning of gutters and the dredging of basins within and around the city Providing and coordinating means of transport and civil engineering
	Ministry of Security and civil protection	• Providing security and relief assistance
Governmental agencies	National Platform for Disaster Risk Reduction	• Gathering risk management information under the early warning system
	National Agency for Civil Protection (ANPC)	Identifying victims, providing assistance and designing awareness messages for the media Providing clean water and evacuation of victims
	National Solidarity Agency (ANC)	Identifying the victims Providing assistance and designing awareness messages
Local Government	Lomé Municipality	Supervision of land use, drainage, flooding areas, and the management retention basins for outflow of rain reduction Providing support in relocation and assistance to victims Designing and implementing the local emergency plan and training the staff
Civil society organisations	International and national NGOs (e.g., OCHA)	Providing financial assistance to victims under the supervision of the municipality Providing food packs, medication, psychological assistance, with support of ANC for disaster intervention (Medical assistance and WASH)

## Data Availability

The data that support the findings of this study are available from the corresponding author. Restrictions apply to the availability of these data, which were used under license for this study. Data are available from the authors with the permission of CSRS.

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
