# Peer review of "Assessing Institutional Stakeholders’ Perception and Limitations on Coping Strategies in Flooding Risk Management in West Africa"

_ijerph, 2022, doi:10.3390/ijerph19116933_

Round 1
Reviewer 1 Report
- As the title is “assessing efficiency of institutional coping strategies”, according to the research, are the institutional coping strategies efficient or not? Actually, this paper is mainly about risk and how to build framework of collaboration between institutions to cope with flood, so “assessing efficiency” is not matching with the content, please think about this?
- As “two different coders from the research team” mentioned in chapter 2.4, what is difference of the “two different coders”, I don’t see in the paper.
- The method of this paper is “Content analysis”, according to the description in chapter 2.4, it is similar to “inductive approach” and “Meta-analysis”, can you emphasize the advantage of “Content analysis” compared with other methods, or why you choose “Content analysis” not “inductive approach” and “Meta-analysis”?
Author Response
# Response to Reviewer 1 Comments
Point 1: As the title is “assessing efficiency of institutional coping strategies”, according to the research, are the institutional coping strategies efficient or not? Actually, this paper is mainly about risk and how to build framework of collaboration between institutions to cope with flood, so “assessing efficiency” is not matching with the content, please think about this?
Response 1: We thank the reviewer for this relevant remark on the manuscript, and we fully agree with this comment. To allow a good coherence of the title and the content, we have made the following changes in the manuscript.
- We have updated the title. The new title is: assessing of institutional stakeholders’ perception and limitations on coping strategies in flooding risk management in West Africa. (Please, see the revised manuscript, lines 1 - 3);
- We have reformulated the study objective in the abstract accordingly to make the content more coherent (lines 16-17);
- We have strengthened the conclusion section (lines 978-986).
We believe that with these changes, the content of the manuscript is in line with the title of the manuscript, the objectives and the conclusion.
Point 2: As “two different coders from the research team ” mentioned in chapter 2.4, what is difference of the “two different coders”?, I don’t see in the paper.
Response 2: Thank you for this remark. We have argued the use of the two coders in this study by adding an explanation sentence (Please, see lines 374-377). Indeed, the use of multi-coder practice enhances the quality of the analysis and increase the confidence in the results as shown by Berends and Jonhston (2009), and gives the opportunity to discuss coding disagreements and refine the coding system.
Point 3: The method of this paper is “Content analysis”, according to the description in chapter 2.4, it is similar to “inductive approach” and “Meta-analysis”, can you emphasize the advantage of “Content analysis” compared with other methods, or why you choose “Content analysis” not “inductive approach” and “Meta-analysis”?.
Response 3: Thank you for this comment about the selection of the Content Analysis (CA) approach in this study. We provided a sentence to explain the relation between CA and the “inductive approach”. Indeed, the “inductive approach” is a category of CA approach, also calls inductive content analysis (Please, see lines 385-387). The meta-analysis in qualitative research is conducted for a rigorous secondary qualitative analysis based on primary qualitative findings, and it is not suitaible for our research. In our manuscript, the inductive approach, as CA category was applied, because it is more adequate for the situation with little literature supporting information, that is the case about the flood management strategies in Abidjan and Togo.

Reviewer 2 Report
The research paper is appropriate and presents a topic of relevant interest to the populations living in Abidjan (Côte d'Ivoire) and Lomé (Togo).
Author Response
# Response to Reviewer 2 Comments
Point 1: The research paper is appropriate and presents a topic of relevant interest to the populations living in Abidjan (Côte d'Ivoire) and Lomé (Togo).
Response 1: We would like to thank the reviewer for this acknowledgement about the quality of our manuscript that show the scientific interest of our manuscript for the populations in Abidjan and Lomé.
Point 2: Are the conclusions supported by the results? Can be improved
Response 2: We thank the reviewer for this comment about the conclusion of our manuscript. We completely agree with the comment on the improvement of the conclusion. We have strengthened the content of our manuscript by improving the title (see, lines 3-4), objective (see lines 17-18), and the conclusion to make it more coherent (Please, see the revised manuscript, lines 983-991).

Round 2
Reviewer 1 Report
Can you explain the different roles of “two different coders from the research team ”, and you sure that “two coders” can be called “multi-coder”?
As you mentioned data collection and data management, where is the data result, I don’t see any data in the whole paper, so which type of data in your paper?
Author Response
Abidjan, May 20, 2022
3rd round of revision for the manuscript: ijerph-1692258
Dear reviewer,
We would like to thank you, for the 3rd round of evaluation of our manuscript entitled “Assessing institutional stakeholders’ perception and limitations on coping strategies in flooding risk management in West Africa”, and for providing us with some major comments.
We have carefully revised our manuscript in light of the points brought forward for consideration. Below, please find our point-by-point response for the reviewers. We have provided answers for each point mentioned.
To further assist you in readily reviewing the changes we have made, we highlighted them in the manuscript with yellow marker and in the response with red marker.
We feel that our manuscript has been further improved thanks to your useful comments, and the revisions we have made based on them.
We enclose the revised version for your consideration and hope that our revisions will meet your expectations, and that you will reconsider our revised manuscript for publication in the International Journal of Environmental Research and Public Health (ijerph).
We would like to thank you and look forward to your reply.
Yours sincerely,
Dr Parfait Kouamé (on behalf of all co-authors)

This manuscript is a resubmission of an earlier submission. The following is a list of the peer review reports and author responses from that submission.
Round 1
Reviewer 1 Report
I thank the authors for the effort to develop this manuscript. My detailed review is attached.

Author Response
Abidjan, March 15, 2022
1st round of revision for the manuscript: ijerph-1607096
Dear reviewer,
We would like to thank you, for the 1st round of evaluation of our manuscript entitled “Assessing efficiency and limitations of institutional coping strategies in flooding risk management in West Africa”, and for providing us with some major comments.
We have carefully revised our manuscript in light of the points brought forward for consideration. Below, please find our point-by-point response. We have provided answers for each point mentioned.
To further assist you in readily reviewing the changes we have made, we highlighted them in the manuscript with blue marker and in the response with yellow marker.
We feel that our manuscript has been further improved thanks to your useful comments, and the revisions we have made based on them.
We enclose the revised version for your consideration and hope that our revisions will meet your expectations, and that you will reconsider our revised manuscript for publication in the International Journal of Environmental Research and Public Health (ijerph).
We would like to thank you and look forward to your reply.
Yours sincerely,
Dr Parfait Kouame (on behalf of all co-authors)
Point-by-point response
ijerph-1607096: (Manuscript Title: "Assessing efficiency and limitations of institutional coping strategies in flooding risk management in West Africa")
We thank you very much for the very useful and detailed comments to improve our manuscript (Please, see attached revised manuscript). We have carefully revised our manuscript according to the reviewers’ comments. Please find our point-by-point response to your comments below. The changes in the word file are visible with track changes, and changes have been marked in green colour.
REVIEWER (R)#2:
General comments: My main concern is lack of data to support results and conclusions which compromises the soundness of the research. Some information seems to be repeated throughout the manuscript and there are several writing errors in the text, figures, tables, and reference list. My recommendation is to reconsider after a major revision. The following comments can help authors to improve their manuscript.
Response R2.1: We thank the reviewer for these important observations, particularly for the matter of data to support results and our conclusions and writing errors. We have addressed these issues highlighted in the revised manuscript (Please, see comments below). For the repeated information, we have carefully revised the whole manuscript to manage errors in the text, figures and tables. Reference list has been reformatted according to MDPI guidelines with Endnote software (Please, see lines 1034-1163).
Comment R2.2: Attempts to discuss similar studies in the study area. Some statements lack references. e.g., lines 37-38, 40-41 and many others. Response R2.1: Thank you for this great suggestion. We have updated the references in the revised manuscript.
Comment R2.2: Introduction tends to generalize Cote d’Ivoire, Togo, west Africa and sub-Saharan Africa. For example, lines 54-57, the paragraph starts with Sub-Saharan Africa, then west Africa and finally cites a study [10] conducted in Ghana. Same observation in lines 66-68 talks about sub-Saharan Africa but cites a study [12] done solely in Niger Delta. I think this may not be correct. I advise that you find studies that are area specific and relate them accordingly. Response R2.2: Thank you for this remark. We completely agree with you about some notes (generalization of the study) making misunderstanding. Indeed, this first paragraph of the introduction highlighted the connection between floods, environment, and health by presenting some literature reviews. We have reformulated sentences by providing literature reviews for sub-Sahara Africa, general West Africa countries and Côte d’Ivoire and Togo (see lines 36-44). We have aggregated the information in this paragraph to make it more understandable (lines 36-71).
Comment R2.3: Please define abbreviations first time on use e.g., SDG line 96 was never defined. Response R2.3: We thank the reviewer for this observation. The abbreviation SDG has been defined in the revised version of the manuscript (see line 98).
Comment R2.4: With exception of figure 1, study area is well described Response R2.4: Thank you for this important comment on figures in the manuscript. Figure 1 has been revised (see lines 172-174)
Comment R2.5: Some information is repeated, and an improvement can be made to reduce the length of the manuscript Response R2.5: Thank you for your insightful comment. We have aggregated some information in the introduction part (see lines 36-71) and in the revised manuscript.
Comment R2.6: Figure 1 should be improved to be in a form of professional plot. There is no description for the larger map the caption and subplot labels A and B should be on their specific plot not just in the caption. My suggestion is that, if no further details are to be shown for Cote d’Ivoire and Togo, no need to isolate the two from the larger map. Please improve the figure borders, add georeferenced ranges of the map, and a compass should be added.
Response R2.6: Many thanks for this observation. Figure has been improved, by adding geoinformation on border in the revised manuscript (see lines 172-174).
Comment R2.7: Why does each figure caption start with the word “near here”? It is a bit confusing. Please correct Figure 1 and 2 captions
Response R2.7: Thank you for your insightful comment. We have removed the word “near here” that was to indicate the place of the figure caption in the submitted version of manuscript (Please, see lines 174 and 217).
Comment R2.8: Both tables have “Table 1” in the caption. Please improve the quality of the first Table 1 and second Table 1. Bullet markers should be well aligned, and bullet points need to be well indented.
Response R2.8: We thank you for these important comments for improving the quality of tables in the manuscript. We have aligned bullet points in the revised version (Table 1: lines 367 - 368, Table 2: lines 370 - 372).
Comment R2.9: No data is presented in the manuscript. Where are data collected during the survey? what type of data was collected and how was it used?
Response R2.9: We would like to thank you for these great comments. In the research design, we clearly indicated that we are conducting a study using a qualitative approach and relying on basis social science methods for data collection and analysis. Data collection and analysis methods mentioned in the revised version of Fig.2 (see line 216) are deeply described in the sections 2.3 on Data collection and 2.4 on Data management and Analysis (see lines 263-306).
Comment R2.10: Which table are you referring to in line 270 since all tables are captioned table 1? Please specify.
Response R2.10: We thank you for this observation. For better understanding this sentence on the line 336, we have rephrased by changing the word “Table 1” by “Tables 1 and 2”.
Comment R2.11: Since tables have important information, both need to be referenced to and discussed in the text, but I see only one table.
Response R2.11: Thank you for this comment. The information on table have been provided and discussed in the text (see lines 577-585).
Comment R2.12: Please remove double numbering on all references i.e 1. [1], 2. [2], ..., etc. I know the editors will also tell you this, but I observe that references list does not meet the journal requirements for references and need to be thoroughly checked. For example, for articles, year of publication should be bold. https://www.mdpi.com/journal/ijerph/instructions
Response R2.12: We thank the reviewer for this important input about references. We have carefully used MDPI_article reference type to format reference in our manuscript as required ijerph. Year of publication is bold (Please, see lines 1034-1163).
Comment R2.13: I recommend that you extensively improve the writing of the entire manuscript. There are several writing errors, repeated statements etc. Just to highlight a few. Line 33 remove “and” before environmental and replace it with comma. Line 62 remove repeated “the”. Line 302, what do words in brackets “[Table 1 near Here, Lomé] mean?
Response R2.13: Thank you, we certainly agree with this great comment. This great suggestion helps to strengthen the issue about English changes to be made. We have addressed these issues in the revised version of the manuscript in the whole manuscript after identifying these small mistakes. English changes have been marked in green color and additional revision in track changes (Please, see to the revised manuscript). Furthermore, in the revised version of our manuscript we have removed these errors “[neat Here, Lomé] (Please, see line 369).

Reviewer 2 Report
Overall, the subject covered by the paper is of particular importance to those regions or communities prone to flood hazards. However, the paper requires a major revision and resubmit. Some comments:
- This review considers the major limitation of the study is that related to the methodological aspect. By looking at Fig. 2, I expected to see, for example, a Content Analysis of the collected data (or similar approaches), however, this is not the case. Therefore, the presented results do not reflect a solid methodological foundation. In fact, why Content Analysis was not considered? Or any other similar approaches?
- By reading the paper, it is unclear which methodology was employed for the analysis. See for example the following (Lines 258-259), “ …In other words, data were classified, sorted and arranged according to specific themes and patterns and develop meaningful conclusions.” Further, References of the employed methodology are not given.
-
One of the approaches that could have been adopted for this sort of analysis, is that related to Content Analysis (CA). CA is a research method which is employed to effectively identify patterns in recorded interviews, speeches, etc. The analysis can also be conducted qualitatively or quantitatively.
-
In general, papers should be explicit in describing which method has been employed for the analysis. One should remember that anyone willing to replicate the results, he/she should be able to do so.
Hence my recommendation is that the authors should be explicit how they reached their results, ie., which method was employed.
Author Response
REVIEWER (R)#1:
Comment R1.1: Moderate English changes required
Response R1.1: Thank you very much for enquiring about this issue about English changes to be made. Based on your recommendations, the manuscript has been thoroughly edited to improve the level of English. Major English changes have been marked in green color (Please, see the revised manuscript).
Comment R1.2: This review considers the major limitation of the study is that related to the methodological aspect. By looking at Fig. 2, I expected to see, for example, a Content Analysis of the collected data (or similar approaches), however, this is not the case. Therefore, the presented results do not reflect a solid methodological foundation. In fact, why Content Analysis was not considered? Or any other similar approaches? By reading the paper, it is unclear which methodology was employed for the analysis. See for example the following (Lines 258-259), “ …In other words, data were classified, sorted and arranged according to specific themes and patterns and develop meaningful conclusions.” Further, References of the employed methodology are not given.
Response R1.2: Thank you very much for this comment and great suggestion to address weaknesses in the methodology section. Even though it was not clearly said in Fig.2 and the methodology section, the study relied basically on Content Analysis for data analysis. Based on your comment, we deemed it necessary to clarify our methodology. Thus, we have modified the Fig.2 to make data collection and analysis processes clearer (see line 216). We also developed a separate section (section 2.4) on Data management and Analysis to clearly explain how data were analysed using the content analysis approach (conventional content analysis in this case). Additionally, we included new references to illustrate the methodological approach used (Please, see references: Hsieh and Shannon, 2005; Elo and Kyngäs 2008; Bengtsson 2016).
Comment R1.3: One of the approaches that could have been adopted for this sort of analysis, is that related to Content Analysis (CA). CA is a research method which is employed to effectively identify patterns in recorded interviews, speeches, etc. The analysis can also be conducted qualitatively or quantitatively.
Response R1.3: Thank you for this important comment to strengthen our research method section. We completely agree with you because CA is a broad method that may be used with either qualitative or quantitative data and in an inductive or deductive (Directed) way. We have addressed this issue in the revised version (see lines 240-253). From the three approaches of CA defined by Hsieh and Shannon (2005) we used the conventional approach to content analysis as compared to directed and summative approaches. The advantage of the conventional CA is that we did not predefine analysis categories from theories and during the analysis, we let the main categories or themes emerging naturally from the analysed data.
Comment R1.5: In general, papers should be explicit in describing which method has been employed for the analysis. One should remember that anyone willing to replicate the results, he/she should be able to do so.
Response R1.5: Thank you for this remark. We fully agree with you, and the method used in the manuscript and data analysis have been addressed in the revised version, as stated in Response R1.2 and Response R1.3. In the research design, we clearly indicated that we are conducting a study using a qualitative approach and relying on basis social science methods for data collection and analysis. Data collection and analysis methods mentioned in the revised version of Fig.2 (see line 216) are deeply described in the sections 2.3 on Data collection and 2.4 on Data management and Analysis (see lines 263-306).
Comment R1.6: Hence my recommendation is that the authors should be explicit how they reached their results, ie., which method was employed
Response R1.6: We thank the reviewer for this important comment. The method section has been improved to address this limitation in the revised manuscript. Detailed information is provided in the section 2.4 on Data analysis to explain how we reached our results (see 263-306).

Round 2
Reviewer 2 Report
The authors did not address any of my comments of my original review.